# Comparative Study on the Models of Thermoreversible Gelation

**DOI:** 10.3390/ijms231810325

**Published:** 2022-09-07

**Authors:** Fumihiko Tanaka

**Affiliations:** Department of Polymer Chemistry, Kyoto University, Kyoto 615-8510, Japan; ftanaka@kmj.biglobe.ne.jp; Tel./Fax: +81-44-988-5440

**Keywords:** thermoreversible gelation, phase separation, cross-linking

## Abstract

A critical survey on the various theoretical models of thermoreversible gelation, such as the droplet model of condensation, associated-particle model, site–bond percolation model, and adhesive hard sphere model, is presented, with a focus on the nature of the phase transition predicted by them. On the basis of the classical tree statistics of gelation, combined with a thermodynamic theory of associating polymer solutions, it is shown that, within the mean-field description, the thermoreversible gelation of polyfunctional molecules is a third-order phase transition analogous to the Bose–Einstein condensation of an ideal Bose gas. It is condensation without surface tension. The osmotic compressibility is continuous, but its derivative with respect to the concentration of the functional molecule reveals a discontinuity at the sol–gel transition point. The width of the discontinuity is directly related to the amplitude of the divergent term in the weight-average molecular weight of the cross-linked three-dimensional polymers. The solution remains homogeneous in the position space, but separates into two phases in the momentum space; particles with finite translational momentum (sol) and a network with zero translational momentum (gel) coexist in a spatially homogeneous state. Experimental methods used to detect the singularity at the sol–gel transition point are suggested.

## 1. Introduction

Thermoreversible gelation in solutions of polymers, as well as of low molecular weight molecules, has been attracting interest from researchers [1,2,3,4,5,6,7]. Many examples of the phase diagrams with sol–gel transition lines have been reported in the literature, early ones of which are atactic polystyrene in various organic solvents [8,9], poly(n-butylmethacrylate) in 2-propanol [10,11], poly(vinyl chloride) in dimethyl malonate [12], etc. Quite recently, some reviews and conceptual works have appeared with relation to responsive gels [13,14], hydrogels for biomedical applications [6,7,15], and hydrogen-bonding supramolecular gelators [16,17,18]. However, the thermodynamic nature of the transition from a sol state to a gel state, and vice versa (sol–gel transition), has not yet been clarified from a scientific point of view. In particular, the fundamental question on whether a real singularity in the physical quantities is accompanied with the transition as is in a usual phase transition, or if the transition is a sharp but continuous crossover, has not been answered yet.

Some of the data reported in the references were obtained by static measurements (differential scanning calorimetry, low-angle light scattering, osmometry, etc.), and others were by rheological measurements (viscosity, complex modulus, diffusion coefficient of test particles, etc.). In a reversible weak gel, cross-links in the network are so weak that they may be broken by an external force. Therefore, the transition point largely depends on the method of the measurement for detection. In addition, the time required to establish gel networks (*gelation time*) is usually much longer than the time taken for the melting of the gel because of the necessary diffusion time for the formation of cross-links by association. Hence, the transition temperature from sol to gel depends on the cooling rate, and is often observed to be lower than that of melting (for cold-setting gels). For these fundamental reasons, there have been intense arguments and serious confusion on the nature of thermoreversible gelation.

In order to study the thermodynamic nature of the sol–gel transition, we must go back to the definition of a gel. The opinion, however, is divided depending on how the researchers observe the gel state. The static measurements listed above attempt to find the *percolation point* where the connectivity of the constituent molecules, or particles, by cross-linking grows to the macroscopic dimensions. At the gel point, the largest cross-linked cluster covers the entire system and loses its center-of-mass translational degree of freedom. The gel point is defined, therefore, by the point where the weight-average molecular weight of the connected clusters (three-dimensional associated polymers) becomes infinite
(1)$${\bar{M}}_{w}=\infty$$
irrespective of whether the bonds, or cross-links, are irreversible or reversible. This definition of the gel point was presented in the classical theory of the gelation reaction [19,20,21,22,23,24].

The gel point can also be defined rheologically as the point where the solution loses its fluidity [3,25]. The diffusion of the molecules in the solution becomes so slow that they look localized on the time scale of the observation. The viscosity is enhanced so highly that the it looks impossible for the solution to flow. The complex viscosity apparently changes from a liquid value to a solid one, and hence the dynamic loss modulus behaves similarly to the storage modulus at the transition point.

These two pictures are mutually complementary. However, to avoid unnecessary confusion, we confine the research in this article to the equilibrium properties, and adopt the first definition Equation (Equation 1) for the gel point. Hence, our solutions may flow after the gel point is passed by the repetition of the dissociation and association of the cross-links, although they retain an entirely connected network structure (*transient gel*).

In this paper, we review various theoretical models of the thermoreversible sol–gel transition reported in the literature and compare them from the viewpoint of statistical thermodynamics. From such a comparative study, we show that the gelation is a special type of condensation by attractive force, i.e., *condensation without surface tension* of the condensate. We also demonstrate that the site–bond correlated percolation model becomes equivalent to the associating polymer solution model in the limit of a strong intermolecular associative force.

## 2. Methods and Results

### 2.1. Condensation without Surface Tension

In order to consider cross-linking by molecular association, we refer back to old studies on the condensation phenomena by a short-range attractive force. We start with the similarity between sol–gel transition and gas–liquid transition. In particular, particulate gels formed by the association of colloid particles have a strong similarity to amorphous liquid particles. To construct the analogy between the gelation of functional molecules in a solvent and condensation of gases into dense phases (liquid or solid), we briefly review the statistical theory of condensation by Mayer and his collaborators [26,27,28,29,30].

To evaluate the partition function of a gas in which molecules interact with the pairwise additive potential u(r), Mayer [26,30] introduced the function
(2)$$f\left(r\right)\equiv e^{-u\left(r\right)/k_{B}T}-1$$
at a given absolute temperature *T* to avoid a divergence due to the repulsive part (hard core) in u(r), and expanded the partition function in powers of the molecular density. The result of Mayer’s theory can be summarized in terms of the power series [31]
(3)$$G_{k}\left(z\right)\equiv \sum_{l\geq 1}l^{k}b_{l}z^{l}$$
for k=0,1,2⋯, whose coefficient $b_{l}$ is the *l*-th cluster integral
(4)$$b_{l}\left(T\right)\equiv \frac{1}{Vl!}\int \cdots \int \sum \prod_{i<j}f\left(r_{i,j}\right)dr_{1}r_{2}\cdots r_{l}$$

The sum is taken over all possible ways of interacting pairs belonging to a single connected cluster. These cluster integrals are formally introduced in the theory to sum up all terms in the density power series. They do not necessarily describe the property of spatially connected molecules (associated molecules), but provide a mathematical tool rather than physical attributes.

The free energy per molecule is
(5)$$F=\ln z-\rho^{-1}G_{0}\left(z\right)$$
in terms of the activity *z*, and the density ρ≡N/V of the molecule ($v\equiv \rho^{-1}$ is the volume per particle). The density and the pressure *p* are given by 
(6a)$$\rho =G_{1}(z)$$
(6b)$$\frac{p}{k_{B}T}=G_{0}(z)$$as functions of the activity. Upon eliminating the activity in these coupled equations, we find the equation of state (*p*–ρ isotherm), through which, we can study phase transitions.

The slope of the pressure in this theory is given by
(7)$${\left(\frac{\partial p/k_{B}T}{\partial \rho }\right)}_{T}=\frac{1}{{\bar{l}}_{w}}={\left(\frac{\partial \ln z}{\partial \ln\rho }\right)}_{T}$$
where
(8)$${\bar{l}}_{w}\equiv \frac{G_{2}\left(z\right)}{G_{1}\left(z\right)}$$
is the weight-average number of molecules in the clusters. The nature of the equation of state is therefore determined by the detailed form of the coefficients $b_{l}$. In particular, the radius of the convergence of the functions $G_{k}\left(z\right)$ and the nature of the singularities on the convergence boundary are governed by the coefficients $b_{l}$ for large *l*.

To study the degree of discontinuity at the gas–liquid transition point, the second derivative of the pressure was also calculated. It is
(9)$${\left(\frac{\partial^{2}p/k_{B}T}{\partial \rho^{2}}\right)}_{T}=\frac{1}{\rho }\left(1-\frac{{\bar{l}}_{z}}{{\bar{l}}_{w}}\right){\left(\frac{\partial \ln z}{\partial \ln\rho }\right)}_{T}$$
where ${\bar{l}}_{z}\equiv G_{3}\left(z\right)/G_{2}\left(z\right)$ is the *z*-average number of molecules in the clusters.

Using the saddle point method, Mayer et al. [28] found that, for a usual force potential of van der Waals type, the cluster integrals take the form
(10)$$b_{l}\left(T\right)≃\frac{\exp[-\gamma \left(T\right)l^{2/3}]}{l^{5/2}}b_{0}{\left(T\right)}^{l}$$
for large *l*, where $b_{0}\left(T\right)$ is a certain nonsingular function of *T*, and γ(T) is the surface tension of the clusters. This asymptotic expression was later derived more rigorously by Born and Fuchs [32] and Kahn and Uhlenbeck [33] by applying the method of steepest descent to the grand partition function as a function of the complex activity. From these studies, the radius of the convergence of the power series $G_{k}\left(z\right)$ is found to be given by $z=b_{0}{\left(T\right)}^{-1}$. The nature of the singular behavior of the pressure and the compressibility depends on whether the series Equation (Equation 3)
(11)$$G_{k}\left(b_{0}^{-1}\right)=\sum_{l=1}^{\infty }\frac{l^{k}}{l^{5/2}}\exp\left[-\gamma \left(T\right)l^{2/3}\right]$$
on the radius of convergence take finite values, or tends to infinity for k=1,2. If the surface tension is positive, these series take finite values for any integer *k*. Hence, there is a finite value
(12)$$\rho_{s}=G_{1}\left(b_{0}^{-1}\right)$$
for the density $\rho_{s}$ at $z=b_{0}^{-1}$. Because Equation ([Disp-formula FD11106-ijms-23-10325]) is divergent (ρ is infinitely large) as soon as *z* exceeds $b_{0}^{-1}$, $\rho_{s}$ is regarded as the density of the condensation point. At this condensation point, the slope of the pressure is finite because $G_{2}\left(b_{0}^{-1}\right)$ in Equation (Equation 6) is finite.

However, if the surface tension γ is zero in a certain temperature range, $G_{0}\left(b_{0}^{-1}\right)=\zeta \left(5/2\right)=1.342$ and $G_{1}\left(b_{0}^{-1}\right)=\zeta \left(3/2\right)=2.612$ are finite, but all higher moments are infinite: $G_{k}\left(b_{0}^{-1}\right)=\infty$ (for k≥2). (ζ(x) is Rieman’s zeta function [34] of *x*.) Therefore, the slope of the pressure is zero at the condensation point.

Mayer assumed the existence of the temperature $T_{m}$ (below the critical temperature $T_{c}$ of a gas–liquid transition), at which, the surface tension of the cluster vanishes. For $T<T_{m}$, the condensation takes place with the finite slope of the pressure (first order phase transition). Because there is a sudden break in the shape of the *p*–ρ curve at $\rho_{s}$, supersaturation of the vapor would be expected. He proved that the existence of the density $\rho_{f}$ (the density of the condensed phase) is larger than $\rho_{s}$, at which, the pressure starts to increase again.

For $T_{m}<T<T_{c}$, the condensation takes place *with zero slope* of the pressure, allowing no extrapolation to a higher pressure of the supersaturated vapor. Compression of the system at a constant temperature in this region induces a uniform increase in density throughout the whole system; droplets change into liquid continuously one by one. He also assumed the existence of the density $\rho_{f}$ (volume $v_{f}$) in this region, but it was a logical guess rather than mathematical proof [29]. The existence of $v_{f}$ can only be confirmed after the structure of the condensed phase (liquid, amorphous solid, crystal, etc.) is clarified. In what follows, we show that thermoreversible gelation is a condensation of the latter type: condensation with no surface tension.

### 2.2. Sponge Phase of Associated Particles

Before moving on to the detailed description of our model of thermoreversible gelation, we next consider the associated particle model of condensing systems proposed by Frenkel [35,36] and Band [37,38,39,40,41]. Just after Mayer’s work appeared, Band [37,38] realized that Mayer’s general results can be derived almost immediately from the theory of associating assemblies discussed by Fowler [31]. In this model, the field of the molecular interaction force is assumed to be limited to a small region around any molecule, so that it forms an assembly of molecules, referred to as *particles*, or *physical clusters*, such as dimers, trimers, etc., in order to distinguish itself from Mayer’s mathematical clusters. However, it neglects interactions among associated particles. The system is therefore regarded as an ideal gas of associated particles. Frenkel [35,36] also introduced the same model to study heterogeneous fluctuations and pre-transition phenomena in the neighborhood of the phase transition.

Consider a system in a gas phase (A) changing into a condensed phase (B) by forming associated particles. Let $N_{A}$ be the number of molecules that remain in the A phase (unimers), and let $N_{l}$ be the number of associated particles consisting of *l* molecules (referred to as *l*-particles) in the B phase. The total number of molecules is given by
(13)$$N=N_{A}+\sum_{l\geq 2}N_{l}$$

The free energy of the system is then found to be
(14)$$F=N_{A}\mu_{A}+\sum_{l\geq 2}\Delta A_{l}N_{l}+k_{B}T\left(N_{A}\ln\frac{N_{A}}{N}+\sum_{l\geq 2}N_{l}\ln\frac{N_{l}}{N}\right)$$
because of the ideality assumption, where $\mu_{A}$ is the chemical potential of the gas phase A, and
(15)$$\Delta A_{l}=\mu_{B}l+\gamma l^{2/3}$$
is the free energy of a particle of the size *l*. Here, $\mu_{B}\equiv \lim_{l\to \infty }\Delta A_{l}/l$ is the chemical potential of a molecule in the B phase, and γ is the surface tension of the particles. The ideal mixing entropy is assumed.

By minimization of this free energy under the constraint (Equation 12), we find that the most probable distribution of the particles takes the form
(16)$$n_{l}\equiv \frac{N_{l}}{N}=e^{-\beta l-\alpha l^{2/3}}$$
where $\beta \equiv \left(\mu_{B}-\mu_{A}\right)/k_{B}T$ is the difference in the chemical potentials, and $\alpha \equiv \gamma /k_{B}T$ is the dimensionless surface tension. The ratio of the number of molecules in the two phases is then given by
(17)$$\frac{N_{B}}{N_{A}}=\sum_{l\geq 2}le^{-\alpha l^{2/3}}{\left[e^{-\left(\mu_{B}-\mu_{A}\right)/k_{B}T}\right]}^{l}$$

The difference from Mayer’s theory $n_{l}=b_{l}z^{l}$ is evident. The pre-factor $l^{-5/2}$ is missing in this treatment. If α>0, the sum is finite at the transition point where $\mu_{A}=\mu_{B}$ holds (and hence $e^{-\left(\mu_{B}-\mu_{A}\right)/k_{B}T}=1$ holds), so the transition is discontinuous. If α=0, there is no phase transition; all molecules continuously grow to particles of the B phase. The nature of the transition is thus directly related to the convergence of the power series.

Hill [42] and Stillinger [43] studied the relation between the Frenkel–Band theory of association equilibrium and Mayer’s cluster integral theory from the viewpoint of intermolecular interaction potential. Stillinger [43] pointed out the possibility of a *sponge-like structure* in physical clusters at temperatures higher than the gas–liquid transition temperature. If the range of the attractive interaction is short enough, thermal agitation would virtually completely overcome the attractive forces. Only the requirement of overlapping particle spheres should hold the clusters together. Such a sponge-like structure can be identified as a gel network, and the transition exactly corresponds to what we will study in the following sections with regard to thermoreversible gelation.

The associated particle model was later refined by Fisher [44] to apply specifically to the droplet formation in a gas–liquid phase transition. Fisher’s droplet model assumes that the associated particle is a droplet with well-defined surface free energy $W_{l}$, which satisfies the condition $W_{l}/l\to 0$ as l→∞, most simply $W_{l}=\gamma \left(T\right)l^{\sigma }$ with 0<σ<1, and neglects interactions between the droplets. The basic equations are the same as Mayer’s Equations ([Disp-formula FD11106-ijms-23-10325]) and ([Disp-formula FD11107-ijms-23-10325]): 
(18a)$$\rho =q_{0}\sum_{l=1}^{\infty }\frac{e^{-\gamma (T)l^{\sigma }}}{l^{\tau -1}}z^{l}$$
(18b)$$\frac{p}{k_{B}T}=q_{0}\sum_{l=1}^{\infty }\frac{e^{-\gamma (T)l^{\sigma }}}{l^{\tau }}z^{l}$$ where $\rho =v^{-1}$ is the density, $q_{0}$ is a constant, σ=1−1/d=2/3 is the exponent of a surface area (d=3 is the space dimension), *z* is the activity of the molecule, and τ(2<τ<2.5) is the parameter due to the closing effect on the surface of the droplet. If γ>0, we have a first-order gas–liquid transition at z=1. At the critical point [44,45], we have the simultaneous conditions $\gamma (T_{c})=0$ and z=1. Hence, $\rho_{c}=q_{0}\zeta \left(\tau -1\right)$ and $p_{c}/k_{B}T_{c}=q_{0}\zeta \left(\tau \right)$, where ζ(x) is the zeta function of *x*.

The droplets model was later explored by many researchers in relation to the critical phenomena [45], nucleation of supersaturated vapor [46], percolation and spinodal [47,48,49], etc. What we consider for gels in this article is the other possibility, such that γ(T)=0, but *z* changes in the region z<1. We therefore have the coupled equations
(19a)$$\rho =q_{0}\sum_{l=1}^{\infty }\frac{l}{l^{\tau }}z^{l}$$
(19b)$$\frac{p}{k_{B}T}=q_{0}\sum_{l=1}^{\infty }\frac{1}{l^{\tau }}z^{l}$$

### 2.3. Associated Particle Model of Polymer Solutions

We now apply the associated particle model to the solutions in which primary functional molecules of the molecular weight *n* (in terms of the number of statistical repeat units on them) carrying the number *f* of functional groups are dissolved in a solvent [4,50,51,52,53,54,55]. We mainly consider polymers with a high *n* value, but can also include low-molecular-weight molecules with small *n*, such as low-mass gelators. For simplicity, we assume that the functionality of the primary molecules is monodisperse and that the functional groups form pairwise bonds that can break and recombine by thermal motion. We consider a model incompressible solution consisting of *N* primary functional molecules and $N_{0}$ solvent molecules. The total volume of the solution is $V=\Omega a^{3}$, where $\Omega \equiv N_{0}+nN$ is the total volume of the solution counted in the unit of the volume $a^{3}$ of a solvent molecule, which is assumed to be equal to the volume of a statistical repeat unit on the primary functional molecule.

Our starting free energy is based on the Flory–Huggins theory of polymer solutions [24,56,57,58], but the molecular association is taken into consideration (referred to as *associating polymer solutions* APS). It is given by
(20)$$F=\sum_{l\geq 1}N_{l}\ln\phi_{l}+N_{0}\ln\left(1-\phi \right)+\sum_{l\geq 1}\beta \Delta A_{l}N_{l}+g\left(\phi \right)N^{G}$$
where $\beta \equiv 1/k_{B}T$, $N_{l}$ is the number of associated particles (physical clusters) formed with *l* primary molecules (referred to as *l*-particle), $\phi_{l}\equiv nN_{l}/\Omega$ is their volume fraction, ϕ≡nN/Ω is the total volume fraction of the primary molecules, and $\Delta A_{l}\equiv \mu_{l}^{\circ }-l\mu_{1}^{\circ }$ is the free energy change accompanying the formation of an *l*-particle from the separate primary molecules in their standard reference state (superscript circle). The mixing enthalpy (Flory’s χ-parameter [56]) need not be considered here because we are not concerned in this paper with the liquid–liquid phase separation induced by the van der Waals interaction. We focus on a reversible gelation by associative force only.

To incorporate the post-gel regime, we have included the last term: the free energy of the gel network (condensed phase) consisting of a macroscopic number $N^{G}$ of the primary molecules. The free energy needed to bind a molecule onto the gel part is given by $g\left(\phi \right)\equiv \beta \left(\mu_{1}^{\circ G}-\mu_{1}^{\circ }\right)$. In general, it depends on the concentration because the structure of the gel changes with the concentration. The specific form of g(ϕ) will be discussed in detail in the following section.

By differentiation, we find, for the chemical potentials
(21a)$$\beta \Delta \mu_{l}=1+\beta \Delta A_{l}+\ln\phi_{l}-nl\rho +nlg^{\prime }(\phi )v^{G}(1-\phi )$$
(21b)$$\beta \Delta \mu_{0}=1+\ln(1-\phi )-\rho -g^{\prime }(\phi )v^{G}\phi$$
(21c)$$\beta \Delta \mu^{G}=-n\rho +g(\phi )+ng^{\prime }(\phi )v^{G}(1-\phi )$$
an *l*-particle, a solvent molecule, and a molecule in the gel, respectively, where
(22)$$\rho \equiv \sum_{l\geq 1}\nu_{l}+1-\phi$$
is the total number of particles possessing the translational degree of freedom, and $g^{\prime }\left(\phi \right)$ is the derivative of g(ϕ). The sum
(23)$$\nu^{S}\equiv \sum_{l\geq 1}\nu_{l}$$
gives the number density of the finite particles in the solution. For the gel part, $\nu^{G}\equiv N^{G}/\Omega$ is the number density of the primary molecules in the gel, and $\phi^{G}=nN^{G}/\Omega =n\nu^{G}$ is the volume fraction of the gel. The gel fraction, defined by the weight fraction of the gel relative to the total weight of the primary molecules, is given by $w=\phi^{G}/\phi$. The volume fraction of the sol part is then $\phi^{S}=\phi \left(1-w\right)$.

In thermal equilibrium, the solution has a distribution of connected particles with a population distribution fixed by the equilibrium condition
(24)$$\Delta \mu_{l}=l\Delta \mu_{1}$$
for association and dissociation. Then, we find that the volume fraction of the *l*-particles is given by
(25)$$\phi_{l}=K_{l}{\phi_{1}}^{l}$$
where $\phi_{1}$ is the volume fraction of the primary molecules that remain free from association (referred to as *unimers*), and $K_{l}=\exp\left(l-1-\beta \Delta A_{l}\right)$ is the equilibrium constant.

Because the volume fraction $\phi_{1}$ of the unimers plays the role of the activity, let us rewrite it as $z\equiv \phi_{1}$. The number $\nu_{l}$ and the volume fraction $\phi_{l}$ are then given by
(26)$$\nu_{l}=b_{l}z^{l},\;\phi_{l}=nlb_{l}z^{l}$$
where the coefficient $b_{l}\equiv K_{l}/nl$ has been introduced. The volume fraction $\phi^{S}$ of the sol part and the total number of associated particles $\nu^{S}$ are then given by
(27a)$$\phi^{S}=\sum_{l\geq 1}\phi_{l}=nG_{1}\left(z\right)$$
(27b)$$v^{S}=\sum_{l\geq 1}v_{l}=G_{0}\left(z\right)$$ where functions
(28)$$G_{k}\left(z\right)\equiv \sum_{l\geq 1}l^{k}b_{l}z^{l}\;\;\left(b_{l}\equiv K_{l}/nl\right)$$
for k=0,1,2⋯ have been introduced. In terms of the gel fraction $w\equiv \phi^{G}/\phi$, the relation
(29)$$\phi =\phi^{S}+\phi^{G}$$
is transformed into the equation for the volume fraction of the sol part
(30)$$\frac{\phi }{n}\left(1-w\right)=G_{1}\left(z\right)$$

In the post-gel regime where the gel exists, there is an additional condition regarding the equilibrium between sol and gel. It is
(31)$$\Delta \mu^{G}=\Delta \mu_{1}$$
which leads to the relation
(32)g(ϕ)=1+lnz

Hence, the binding free energy g(ϕ) is uniquely related to the activity *z* of the functional molecule.

The osmotic pressure of the solution is related to the chemical potential of the solvent by the thermodynamic relation $\pi a^{3}/k_{B}T=-\Delta \mu_{0}$. Explicitly, we have
(33)$$\frac{\pi a^{3}}{k_{B}T}=\nu^{S}-\phi -\ln\left(1-\phi \right)+\frac{1}{n}g^{\prime }\left(\phi \right)\phi w$$

In the pre-gel regime, the pressure is basically proportional to the total number $\nu^{S}$ of associated particles, since all molecules with a translational degree of freedom equally contribute to the pressure within the ideal gas approximation. In the post-gel regime, there is a contribution from the gel given by the last term. If the binding energy is independent of the concentration, however, the osmotic pressure is independent of the gel fraction. By solving Equation (Equation 25) with respect to *z*, and substituting the result into Equation (Equation 28), we obtain the osmotic pressure as a function of the temperature and volume fraction of the primary molecules.

The total free energy per primary molecule is
(34)$$F=1+\ln z-\frac{1}{\phi }G_{0}\left(z\right)+\frac{n}{\phi }\left(1-\phi \right)\ln\left(1-\phi \right)$$

This is analogous to Mayer’s formula (Equation 5), except the last term (mixing entropy of the solvent).

We next calculate the slope of the osmotic pressure ∂π/∂ϕ, and find the osmotic compressibility defined by $K_{T}\equiv \left(k_{B}T/a^{3}\right){\left(\partial \phi /\partial \pi \right)}_{T}/\phi$ as a function of the temperature and the volume fraction. If the pressure slope becomes zero, the compressibility is divergent. Thus, it indicates a phase transition.

By taking the concentration derivative of Equation (Equation 28), we find
(35)$${\left(\frac{\partial \pi a^{3}/k_{B}T}{\phi \partial \phi }\right)}_{T}=\frac{\kappa (z)}{n\phi }+\frac{1}{1-\phi }$$
for the slope of the osmotic pressure. Here, the new factor κ is defined by
(36)$$\kappa \left(z\right)\equiv n{\left(\frac{\partial \nu^{S}}{\partial \phi }\right)}_{T}=\frac{G_{1}\left(z\right)}{G_{2}\left(z\right)}=\frac{1}{{\bar{l}}_{w}\left(z\right)}$$
in the pre-gel regime, where ${\bar{l}}_{w}\left(z\right)$ is the weight-average degree of polymerization of cross-linked polymers. The activity *z* is a function of ϕ and temperature *T* through the relation ([Disp-formula FD11255-ijms-23-10325]). Since we have, alternatively,
(37)$$\kappa \left(z\right)={\left(\frac{\partial \ln z}{\partial \ln\phi }\right)}_{T}$$
this result is the same as Mayer’s one (Equation 6), except the last term 1/(1−ϕ) in the slope (Equation 30), which comes from the mixing entropy of solvent molecules. (In the case of gas–liquid transition, vacancy plays the role of the solvent and has no mixing entropy.) This term does not cause any singularity. It simply gives the increment of the pressure in the high concentration region of the primary molecules due to the finiteness of the molecular volume. If we use the approximation ln(1−ϕ)≃−ϕ in the last term of Equation (Equation 29) by assuming a low concentration, and put n=1 for low-molecular-weight molecules, Mayer’s formula is exactly recovered.

In the post-gel regime, the function κ(z) in the slope has an additional term due to the finite gel fraction
(38)$$\kappa \left(z\right)\equiv \frac{\partial }{\partial \ln\phi }\left(1+w\frac{\partial }{\partial \ln\phi }\right)\ln z$$

The singularity in the osmotic pressure originates in this κ function: the translational entropy of associated particles.

At this stage, we introduce a new function σ(ϕ,T) by the definition
(39)$$\sigma \left(\phi ,T\right)\equiv {\left(\frac{\partial \pi a^{3}/k_{B}T}{\phi \partial \phi }\right)}_{T}=\frac{\kappa (z)}{n\phi }+\frac{1}{1-\phi }$$
for later convenience. The osmotic compressibility is described by $K_{T}^{-1}=\phi^{2}\sigma \left(\phi ,T\right)$, so that the spinodal condition is simply
(40)σ(ϕ,T)=0

If we included the enthalpy of mixing (van der Waals interaction) in terms of Flory’s χ-parameter in our starting free energy of the APS model, we would have obtained
(41)$$\sigma \left(\phi ,T\right)=\frac{\kappa (z)}{n\phi }+\frac{1}{1-\phi }-2\chi \left(T\right)=0$$
for the spinodal condition.

### 2.4. Application of the Classical Gelation Theory

We now consider specific models for association [4]. We first split the free energy of association into three parts: $\Delta A_{l}=\Delta A_{l}^{\mathrm{comb}}+\Delta A_{l}^{\mathrm{conf}}+\Delta A_{l}^{\mathrm{bond}}.$

In order to find the combinatorial part $\Delta A_{l}^{\mathrm{comb}}$, all particles are assumed to take tree forms. The cycle formation within a particle is neglected. We consider the entropy change in combining *l* identical *f*-functional molecules into a single Cayley tree. The classical tree statistics [22,23] (see also Chapter XII in Flory’s textbook [24]) gives $\Delta S_{l}^{\mathrm{comb}}=k_{B}\ln\left[f^{l}\omega_{l}\right]$ for the entropy of combination, where
(42)$$\omega_{l}\equiv \frac{(fl-l)!}{l!(fl-2l+2)!}$$
is Stockmayer’s combinatorial factor. The free energy is given by $\beta \Delta A_{l}^{\mathrm{comb}}=-\Delta S_{l}^{\mathrm{comb}}/k_{B}$.

For the conformational free energy $\Delta A_{l}^{\mathrm{conf}}$, we employ the lattice theoretical entropy of disorientation [24,56]
(43)$$S_{\mathrm{dis}}\left(n\right)=k_{B}\ln\left(\frac{n\zeta {\left(\zeta -1\right)}^{n-2}}{\sigma_{s}e^{n-1}}\right)$$
for a chain consisting of *n* statistical units, where ζ is the lattice coordination number, and $\sigma_{s}$ the symmetry number of the chain. This entropy is produced when a polymer chain carrying the number *n* of the statistical units is brought from the hypothetical crystalline state to an amorphous one. The first bond starting from the chain end can be placed in any direction among the nearest-neighboring ζ cells. The following bonds have only ζ−1 possible ways of placement because one of the nearest-neighboring cells is already occupied by the preceding monomer. We thus have the factor ${\left(\zeta -1\right)}^{n-2}$. The remaining factor $n/\sigma_{s}e^{n-1}$ is the artifact of the lattice theory. We then find
(44)$$\Delta S_{l}^{\mathrm{conf}}=S_{\mathrm{dis}}\left(ln\right)-lS_{\mathrm{dis}}\left(n\right)=k_{B}\ln\left[{\left(\frac{\sigma_{s}{\left(\zeta -1\right)}^{2}}{\zeta en}\right)}^{l-1}l\right]$$

Finally, the free energy of bond formation $\Delta A_{l}^{\mathrm{bond}}$ is given by
(45)$$\beta \Delta A_{l}^{\mathrm{bond}}=\left(l-1\right)\beta \Delta f_{0}$$
because there are l−1 bonds in a tree of *l* molecules ($\Delta f_{0}$ is the free energy change on forming one bond).

Combining all of the results together, we find
(46)$$K_{l}=nfl\omega_{l}{\left(\frac{f\lambda }{n}\right)}^{l-1}$$
for the equilibrium constant, where
(47)$$\lambda \left(T\right)\equiv \frac{\sigma_{s}{\left(\zeta -1\right)}^{2}}{\zeta e}e^{-\beta \Delta f_{0}}$$
is the *association constant*, which provides a measure of the strength of a physical bond.

The total volume fraction and the total number of particles in the solution are then given by using Equations ([Disp-formula FD11255-ijms-23-10325]) and ([Disp-formula FD11256-ijms-23-10325]) as
(48a)$$\frac{\lambda }{n}\phi (1-w)=\sum_{l=1}^{\infty }l\omega_{l}z^{l}$$
(48b)$$\lambda v^{S}=\sum_{l=1}^{\infty }\omega_{l}z^{l}$$


The parameter *z* is here defined by $z\equiv \lambda f\phi_{1}/n$. ($\phi_{1}$ being the volume fraction of the unimers.)

#### 2.4.1. Pre-Gel Regime

We first consider the pre-gel regime where all particles are finite. From the fundamental two relations ([Disp-formula FD12223-ijms-23-10325]) and ([Disp-formula FD12224-ijms-23-10325]) given above with w=0, the function κ is given by
(49)$$\kappa \left(z\right)=\frac{\sum l\omega_{l}z^{l}}{\sum l^{2}\omega_{l}z^{l}}=\frac{1}{{\bar{l}}_{w}\left(z\right)}$$
so that it is identified to be the reciprocal of the weight-average aggregation number ${\bar{l}}_{w}$ of particles. In what follows, we show that κ is continuous at the gel point concentration, but its derivative ${\left(\partial \kappa /\partial \phi \right)}_{T}$ has a discontinuity.

It is now clear that the moments of the Stockmayer’s distribution defined by
(50)$$S_{k}\left(z\right)\equiv \sum_{l=1}^{\infty }l^{k}\omega_{l}z^{l}\;\left(k=0,1,2,\cdots \right)$$
play exactly the same roles of the functions $G_{k}\left(z\right)$ in the theories of Mayer and Frenkel–Band. The number-average and weight-average of the particle distribution are given by
(51a)$${\bar{l}}_{n}=\frac{S_{1}\left(z\right)}{S_{0}\left(z\right)}$$
(51b)$${\bar{l}}_{w}=\frac{S_{2}\left(z\right)}{S_{1}\left(z\right)}$$

The osmotic pressure is
(52)$$\frac{\pi a^{3}}{k_{B}T}=\frac{S_{0}\left(z\right)}{\lambda (T)}-\phi -\ln\left(1-\phi \right)$$
where the activity *z* is related to the volume fraction ϕ by
(53)$$\frac{\lambda \phi }{n}=S_{1}\left(z\right)$$

The slope of the pressure is
(54)$${\left(\frac{\partial \pi a^{3}/k_{B}T}{\phi \partial \phi }\right)}_{T}=\frac{1}{n\phi {\bar{l}}_{w}\left(z\right)}+\frac{1}{1-\phi }$$

The moments $S_{k}\left(z\right)$ can be exactly calculated by introducing a new parameter α, which is the positive root of the equation
(55)$$z\equiv \alpha {\left(1-\alpha \right)}^{f-2}$$
for a given value of *z*. The first three moments of Stockmayer’s distribution are explicitly calculated to be [22]
(56a)$$S_{0}\left(z\right)=\frac{\alpha (1-f\alpha /2)}{f{\left(1-\alpha \right)}^{2}}$$
(56b)$$S_{1}\left(z\right)=\frac{\alpha }{f{\left(1-\alpha \right)}^{2}}$$
(56c)$$S_{2}\left(z\right)=\frac{\alpha (1+\alpha )}{f[1-(f-1)\alpha ]{\left(1-\alpha \right)}^{2}}$$

In order to see the physical meaning of α, let us calculate the extent of the reaction, i.e., the probability for a randomly chosen functional group to be associated. Since an *l*-particle includes the total of fl functional groups, among which, 2(l−1) are associated, the extent of the reaction is given by $2\left[S_{1}\left(z\right)-S_{0}\left(z\right)\right]/fS_{1}\left(z\right)=\alpha .$ Thus, it turns out that α, introduced by the formal relation (Equation 49), actually gives the extent of the reaction.

By using α, the average particle sizes are given by
(57a)$${\bar{l}}_{n}=\frac{1}{1-f\alpha /2}$$
(57b)$${\bar{l}}_{w}=\frac{1+\alpha }{1-(f-1)\alpha }$$

Therefore, it is obvious that the gel point is identified to be $\alpha =\alpha^{*}$, where
(58)$$\alpha^{*}\equiv \frac{1}{f-1}$$
because the weight-average particle size becomes infinite at this point. The activity at the gel point is then given by
(59)$$z^{*}\equiv \frac{{\left(f-2\right)}^{f-2}}{{\left(f-1\right)}^{f-1}}$$

The volume fraction at the gel point is therefore
(60)$$\frac{\lambda \phi^{*}}{n}=S_{1}\left(z^{*}\right)=\frac{\left(f-1\right)}{f{\left(f-2\right)}^{2}}$$

All moments are monotonically increasing functions of *z*, and have a common radius of convergence $z^{*}$. For $z>z^{*}$, all moments diverge. Exactly on the radius of convergence $z^{*}$, $S_{0}$ and $S_{1}$ take finite values, but all moments with k≥2 are infinite. The number average also diverges at $\alpha =\alpha_{0}\equiv 2/f$, but, since $\alpha^{*}<\alpha_{0}$, we have to study its post-gel behavior on the basis of the proposed treatment of the post-gel regime.

In terms of the reactivity α, we find the osmotic pressure as
(61)$$\frac{\pi a^{3}}{k_{B}T}=\left(\frac{1}{n}-1\right)\phi -\frac{f}{2n}\phi \alpha -\ln\left(1-\phi \right)$$

The slope of the pressure is
(62)$${\left(\frac{\partial \pi a^{3}/k_{B}T}{\phi \partial \phi }\right)}_{T}=\frac{1-(f-1)\alpha }{n\phi (1+\alpha )}+\frac{1}{1-\phi }$$

In the pre-gel regime ($\alpha <\alpha^{*}$), the volume fraction $\phi^{S}$ occupied by the molecules belonging to the sol must always be equal the total polymer volume fraction ϕ. Thus, from Equation ([Disp-formula FD12223-ijms-23-10325]), the total volume fraction ϕ and the extent of association α satisfy the relation
(63)$$\lambda \psi =\frac{\alpha }{{\left(1-\alpha \right)}^{2}}$$
where ψ≡fϕ/n (the total number concentration of the functional groups) is used instead of the volume fraction ϕ.

We can solve this equation for α, and find
(64)$$\alpha =\frac{1}{2\lambda \psi }\left\{1+2\lambda \psi -\sqrt{1+4\lambda \psi }\right\}$$

At the sol–gel transition point, the binding free energy, the free energy for a primary molecule to be bound to the gel network, is
(65)$$g\left(\phi^{*}\right)=1+\left(f-1\right)\ln\left(f-1\right)-\left(f-2\right)\ln\left(f-2\right)+\ln\left(f\lambda /n\right)$$
from Equation (Equation 27). This critical value gives
(66)$$z^{*}=\frac{n}{f\lambda }\exp\left[g\left(\phi^{*}\right)-1\right]$$
which agrees with Equation (Equation 51). The concentration of polymers at the gel point is then given by Equation (Equation 52), or
(67)$$\lambda \left(T\right)\psi^{*}=\frac{f-1}{{\left(f-2\right)}^{2}}$$

This condition gives the sol–gel transition line on the temperature-concentration plane.

To study the singularity at the gel point, we examine how the weight-average diverges. A simple calculation gives
(68)$${\bar{l}}_{w}\left(z\right)≃\frac{A}{\phi^{*}-\phi }$$
and hence
(69)$$\kappa \left(z\right)=\frac{\phi^{*}-\phi }{A}$$
with the amplitude
(70)$$A\equiv \frac{fn}{{\left(f-2\right)}^{3}\lambda \left(T\right)}$$

The analogy between vapor condensation and gelation in the random polymerization of polyfunctional molecules was originally noticed by Stockmayer [22] in his statistical–mechanical analysis of the gelation reaction and molecular weight distribution function. Later, the analogy was explored by introducing physical clusters rather than Mayer’s mathematical clusters [42,43]. On the basis of Hill’s criterion for a bond formation, an attempt to derive Stockmayer distribution within the theoretical framework of Mayer was made under the condition of no ring-closure [59,60]. Gibbs et al. [61] applied the tree approximation to the Mayer’s cluster integrals and derived very flat *p*–*v* isotherms, the horizontal portion of which represents gas–liquid coexistence. The form that they assumed for the cluster integrals was
(71)$$b_{l}=\frac{l^{l-2}}{l!}b_{0}{\left(T\right)}^{l-1}$$
so that it is very close to the asymptotic form of Equation (Equation 9), but leads to slightly different behaviors of $G_{k}\left(z\right)$ near the transition point.

#### 2.4.2. Similarity to Bose–Einstein Condensation

At this stage, we readily realize that Equations ([Disp-formula FD12223-ijms-23-10325]) and ([Disp-formula FD12224-ijms-23-10325]) are mathematically parallel to those we encountered in the study of the Bose–Einstein condensation (BEC) of ideal Bose gases [30,62,63,64]. The number density N/V and the pressure *p* of an ideal Bose gas consisting of *N* molecules confined in the volume *V* is given by
(72a)$$\frac{\lambda_{T}^{3}N}{V}=\sum_{l=1}^{\infty }\frac{z^{l}}{l^{3/2}}$$
(72b)$$\frac{\lambda_{T}^{3}p}{k_{B}T}=\sum_{l=1}^{\infty }\frac{z^{l}}{l^{5/2}}$$
where *z* is the activity of the molecule, and $\lambda_{T}\equiv h/{\left(2\pi mk_{B}T\right)}^{1/2}$ is the thermal de Broglie wave length. The coefficient of the infinite series on the right hand side is replaced from Stockmayer’s combinatorial factor $\omega_{l}$ to $1/l^{5/2}$, but other parts are completely analogous. The infinite summations on the right hand side of these equations are known as Truesdell’s function [65] of order 3/2 and 5/2. Their singularity appearing at the convergence radius z=1 was studied in detail [65]. Since the internal energy of the Bose gas is related to the pressure as U=3pV/2, the singularity in the compressibility and that in the specific heat have the same nature and reveal discontinuity in their derivatives [62,63,64]. (See also Chapter 14 in Mayer’s textbook [30].) Hence, the transition (condensation of macroscopic number of molecules into a single quantum state of zero momentum) turns out to be a third-order phase transition.

We now show that a similar picture holds for our gelling solution; a finite fraction of the total number of primary molecules condenses into a single state (gel network) with no center of mass translational degree of freedom (no momentum), although we have no quantum effect. The gel network extends to the entire system, and hence loses its translational degree of freedom.

The analogy of BEC can be seen more clearly if we replace Stockmayer’s combinatorial factor $\omega_{l}$ by its asymptotic form
(73)$$\omega_{l}≃\frac{1}{l^{5/2}}{\left(\frac{1}{z^{*}}\right)}^{l}$$
for large *l*, where $z^{*}$ is the critical value (Equation 51). This form can be derived by applying Stirling’s formula $N!≃\sqrt{2\pi N}N^{N}e^{N}$ for a large *N* to Equation (Equation 37). We find that
(74a)$$\frac{\lambda \phi }{n}=\sum_{l=1}^{\infty }\frac{1}{l^{3/2}}{\left(\frac{z}{z^{*}}\right)}^{l}$$
(74b)$$\lambda v=\sum_{l=1}^{\infty }\frac{1}{l^{5/2}}{\left(\frac{z}{z^{*}}\right)}^{l}$$

Thus, we can see that the singularity at $z=z^{*}$ is identical to those in Truesdell’s functions at z=1. In our previous study [54], we referred to this important similarity in the more general case of multiple association. Later, the nature of the singularity was clarified [66] in relation to the coexisting phase separation.

Comparing with the asymptotic form of Equation (Equation 9) of the cluster integral, we can readily see that there is no term corresponding to the surface tension in $\omega_{l}$ for the tree particles. The reason why particles of the tree form have no surface tension is easily understood as follows. The surface free energy of a particle is given by $\gamma \left(T\right)l^{(d-1)/d}$ in a space of dimensions *d*. Because the dimensions of a tree are infinite [67], the surface free energy is proportional to the size *l* of the tree, so that it can be adsorbed into the factor $z^{*}$.

Physically, a primary molecule on the surface of an associated particle of tree form has the same, or with negligible difference, contact number with solvent molecules as that of a free primary molecule because of the geometrical characteristics of a tree form. Therefore, no significant difference in the interaction energy is produced when a primary molecule is attached on a tree particle, thus leading to no surface tension.

We can push this analogy further still by considering loops (rings) instead of trees. To overcome the severe limitation in accounting for cyclic structures in tree statistics, Jacobson and Stockmayer [68,69] studied the linear polycondensation (f=2) in which open chains and closed loops coexist. If we apply their idea to reversible loop formation, we can treat it within the theoretical framework of Mayer, Frenkel–Band, or APS. The cluster integrals for a loop of size *l* is exactly given by
(75)$$b_{l}=\frac{1}{l^{5/2}}b_{0}{\left(T\right)}^{l}$$
because the probability of closing the ends of an open chain constructed with the number *l* of constituent Gaussian chains is proportional to $1/l^{5/2}$ (including the symmetry number). If the primary chains are not Gaussian, but obey the scaling law due to the excluded volume effect, the ring closure probability is proportional to $1/l^{\tau }$, where τ=νd+γ−1. (d=3 is the space dimensions, ν=0.6 is the Flory’s exponent of the radius of gyration of a chain, and γ=1.13 is the exponent of the total number of self-avoiding random walks [70].) The exponent τ changes from 2.5 to 2.96, but the nature of the functions $G_{k}\left(z\right)$ ($G_{0},G_{1}$ are finite whereas $G_{k}\;\left(k\geq 2\right)$ are infinite at z=1) remains the same, so the singular behavior of the osmotic pressure remains the same [71].

A similar factor of the loop entropy appeared in the fusion of DNA double helices [72,73,74]. When a double helix melts partially, a loop made up of the combination of the complementary strands is created, and the entropy of loop formation appears. Thus, the BEC of loops is exactly reproduced within the classical statistical mechanics [71,72,73,74].

#### 2.4.3. Analyses of the Singularity

In the post-gel regime where $\phi >\phi^{*}$ ($\alpha >\alpha^{*}$), we have an additional equilibrium condition (Equation 27). The activity *z* of the solute molecule is related to its binding free energy of the gel.

Since the reactivity of a functional (associative) group in the sol can, in general, be different from that in the gel, let us write the former as $\alpha^{S}$ and the latter as $\alpha^{G}$. The average reactivity α of the system as a whole is given by
(76)$$\alpha =\alpha^{S}\left(1-w\right)+\alpha^{G}w$$
where *w* is the weight fraction of the gel.

The volume fraction $\phi^{S}=\phi \left(1-w\right)$ of polymers belonging to the sol is consequently given by
(77)$$\frac{\lambda \phi^{S}}{n}=S_{1}\left(\alpha^{S}\right)$$
in the post-gel regime, so it is different from the total ϕ that is given by $S_{1}\left(\alpha \right)$. The total number of finite clusters must also be replaced by
(78)$$\lambda \nu^{S}=S_{0}\left(\alpha^{S}\right)$$

This gives the number of particles that have a translational degree of freedom. The gel network covers the entire solution and has no translational degree of freedom. If we use the total reactivity α in this equation instead of $\alpha^{S}$ in the post-gel regime, we have an unphysical result such that $\nu^{S}$ becomes negative for α>2/f because of Equation ([Disp-formula FD11156-ijms-23-10325]).

The problem regarding how the reaction inside the gel proceeds in the post-gel regime depends on the structure of the gel. To find $\alpha^{G}$ as a function of the concentration of the primary functional molecules, several models are possible.

#### 2.4.4. Stockmayer’s Treatment

Because all particles are assumed to take a tree form, Stockmayer [22,23] proposed that the gel must also retain a tree form. Hence, in the post-gel regime, the extent of the reaction in the gel should take the value $\alpha^{G}=\alpha_{0}$, where
(79)$$\alpha_{0}\equiv \lim_{l\to \infty }\frac{(f-2)l+2}{fl}=\frac{2}{f}$$
is the reactivity of an infinite tree structure without cycles. He also assumed that the extent of the reaction of functional groups in the finite particles remains at the critical value $\alpha^{S}=\alpha^{*}$ throughout the post-gel regime. The osmotic pressure is therefore given by
(80)$$\frac{\pi a^{3}}{k_{B}T}=\frac{S_{0}\left(z^{*}\right)}{\lambda (T)}-\phi -\ln\left(1-\phi \right)$$

From the definition (Equation 66), the gel fraction *w* takes the form
(81)$$w=\frac{(f-1)\alpha -1}{1-\alpha_{0}}$$
where α ($>\alpha^{*})$ is the extent of the reaction of the entire system, including all functional groups. It is a linear function of α, and reaches unity (all molecules belong to the gel) at $\alpha_{0}$, *before the reaction is completed*. The volume fraction of the sol remains constant at $\phi^{S}=\phi^{*}$. The number-average particle size remains constant at ${\bar{l}}_{n}=\left(f-2\right)/2\left(f-1\right)$, whereas the weight-average is divergent ${\bar{l}}_{w}=\infty$ in the post-gel regime.

In Figure 1, the osmotic pressure $\pi a^{3}/k_{B}T$ as functions of the volume fraction of primary molecules is shown. For simplicity, the primary molecules are assumed to be trifunctional f=3 and of low molecular weight (n=1). The association constant λ is changed from curve to curve. The gel point is indicated by a circle for each value of λ. The pressure is continuous at the gel point.

In Figure 2, the compressibility for the same systems is shown. At the gel point, it is continuous, but has a cusp whose slope is discontinuous. This slope discontinuity is enhanced by the increase in the association constant λ. The line connecting the top of the cusps forms a sol–gel transition line.

We thus find that the discontinuity in the slope of the function κ is given by
(82)$$\Delta {\left(\frac{\partial \kappa }{\partial \phi }\right)}_{T}=\frac{1}{A}$$

This leads to a discontinuity in the osmotic compressibility of the form
(83)$$\Delta {\left(\frac{\partial K_{T}}{\partial \phi }\right)}_{T}=-K_{T}^{2}\left(\frac{\phi^{*}}{n}\right)\Delta {\left(\frac{\partial \kappa }{\partial \phi }\right)}_{T}=-\frac{B}{\sigma {\left(\phi^{*},T\right)}^{2}}$$
where
(84)$$B\equiv \frac{f^{2}{\left(f-2\right)}^{9}\lambda {\left(T\right)}^{4}}{{\left(f-1\right)}^{3}n^{5}}$$
is a constant depending only on the temperature, functionality, and the number of statistical units on a chain. For large-molecular-weight primary molecules, the amplitude *B* is small. This is the main reason why the experimental detection of the singularity has so far been difficult.

The binding free energy is constant at the value of Equation (Equation 57). From Equation ([Disp-formula FD12223-ijms-23-10325]), which is now equivalent to
(85)$$\lambda \psi \left(1-w\right)=\lambda \psi^{*}$$
we find
(86)$$w=1-\phi^{*}/\phi$$

The result is schematically shown in Figure 3 on the temperature–concentration phase plane. If we cool the solution from P at a constant concentration, we hit the sol–gel transition line at temperature $T_{g}$, where the gel network starts to appear. The gel extends to the entire system without phase separation in the coordination space. The molecules are however separated into two phases in the momentum space—molecules with finite momentum (sol) and with zero momentum (gel)—because the gel is macroscopic and has no degree of freedom for the center of mass translational motion. The concentration of the sol remains at the critical value, and therefore the solution moves along the sol–gel transition line by further cooling. The gel grows, but retains a tree structure.

#### 2.4.5. Flory’s Treatment

Theoretically, Stockmayer’s picture [22,23] is not the only consistent way to treat the post-gel regime, but other pictures are possible without breaching the fundamental laws of thermodynamics. In fact, Flory proposed a different picture in his work on the gelation reaction of trifunctional molecules [19,20,21]. In his treatment, molecules in the sol part react with those in the gel part with an increase in the concentration, and, as a result, the formation of *cyclic linkages within the gel part* is allowed.

Using the definition (Equation 49) for α, the activity *z* takes a maximum value $z=z^{*}$ at $\alpha =\alpha^{*}$. Therefore, two values of α can be found for a given value of *z* in the post-gel regime, where $\alpha >\alpha^{*}$ holds. For a given α, the value of *z* is fixed by the relation (Equation 49). There is another root $\alpha^{\prime }$ (shadow root) lying below $\alpha^{*}$ of the equation for a given value of *z*. Flory postulated that $\alpha^{\prime }$ gives the extent of the reaction in the sol. Hence, we have
(87)$$\alpha^{S}=\alpha^{\prime }$$

For the total volume fraction, the relation
(88)$$\frac{\lambda \phi }{n}=S_{1}\left(\alpha \right)$$
remains valid. The volume fraction of the sol part in Equation (Equation 67) is changed to
(89)$$\frac{\lambda \phi^{S}}{n}=S_{1}\left(\alpha^{\prime }\right)$$
leading to the sol fraction $w^{S}=S_{1}\left(\alpha^{\prime }\right)/S_{1}\left(\alpha \right)$. Equation (Equation 68) is also changed to
(90)$$\lambda \nu^{S}=S_{0}\left(\alpha^{\prime }\right)$$
because the number of particles can be counted only for finite particles In the literature [75], there is a statement that “In fact one can easily check that the free energy, eqn (2.15), and all its derivatives are perfectly analytical at the gel point”. The error in this paper resides in the treatment of the reaction in the postgel regime. It claims that the study is based on Flory’s postgel picture, but in fact it simply missed the extent of reaction $\alpha^{\prime }$ of the sol, and hence it failed to find the gel fraction. (Superscript S indicates the sol part.)

The function κ that appeared in the compressibility is
(91)$$\kappa =n{\left(\frac{\partial \nu^{S}}{\partial \phi }\right)}_{T}=\frac{1}{{\bar{l}}_{w}^{S}\left(\alpha^{\prime }\right)}$$
where
(92)$${\bar{l}}_{w}^{S}\left(\alpha^{\prime }\right)=\frac{1+\alpha^{\prime }}{1-\left(f-1\right)\alpha^{\prime }}$$
refers to the weight-average particle size in the sol part in the post-gel regime. Thus, from the post-gel form of Equation (Equation 33) of κ, we also find a discontinuity
(93)$$\Delta {\left(\frac{\partial \kappa }{\partial \phi }\right)}_{T}=-\frac{4}{A}$$
in the slope of the osmotic compressibility in Flory’s treatment, although the sign of the discontinuity becomes negative. The gel fraction is given by
(94)$$w=1-\frac{{\left(1-\alpha \right)}^{2}\alpha^{\prime }}{{\left(1-\alpha^{\prime }\right)}^{2}\alpha }$$

The gel fraction reaches unity only at the limit of complete reaction α=1. The extent of association $\alpha^{G}$ in the gel can be obtained by the definition of the total reactivity (Equation 66). Explicitly, it gives
(95)$$\alpha^{G}=\frac{\alpha +\alpha^{\prime }-2\alpha \alpha^{\prime }}{1-\alpha \alpha^{\prime }}$$

This value is obviously larger than $\alpha_{0}$ (infinite limit of the tree) so that, in Flory’s picture, *cycle formation is allowed within the gel network*. Its cycle rank is given by
(96)$$\xi =\frac{\alpha^{G}}{2}-1$$

The free energy g(ϕ) for binding a primary molecule onto the gel network turns out to be
(97)$$g\left(\phi \right)=1-\left(f-1\right)\ln\left(\lambda \psi \right)+f\ln[\left(\sqrt{1+4\lambda \psi }-1\right)/2]$$

It is a monotonically decreasing function of the concentration. With an increase in the concentration, the network structure becomes tighter, so the binding of a polymer chain becomes stronger. Since the average number of bonds per molecule is $\left(f/2\right)\alpha^{G}=\alpha^{G}/\alpha_{0}$, the binding free energy *per bond* is given by $\alpha_{0}g\left(\phi \right)/\alpha^{G}$. This is not a constant, but changes as the reaction proceeds.

The result is schematically shown in Figure 4. If we cool the solution from P at a constant concentration, we hit the sol–gel transition line at temperature $T_{g}$, where the gel network starts to appear. By further cooling, the solution moves along the new line, which lies above the sol–gel transition line because the concentration of the sol decreases below the critical value. The gel network grows and forms cycles within it.

#### 2.4.6. Other Post-Gel Treatments

The difference in the above two treatments was later examined from a kinetic point of view. For an irreversible cross-linking reaction, Ziff and Stell [76] clarified the reaction mechanism (sol–gel interaction) in these two treatments after the gel point is passed. They found that, in Stockmayer’s treatment, reactive groups in the sol do not interact with those in the gel, and the gel grows only through a *cascade process* from the sol to the gel, whereas, in Flory’s treatment, all functional groups are allowed to react. On the basis of their kinetic study, they proposed a third model that takes the reaction between the sol and gel into account as in Flory’s picture, while the cycle formation in the gel is forbidden as in Stockmayer’s one.

In classical tree statistics, the number of the functional groups on the surface of a tree-like cluster is of the same order as that of the groups inside the cluster, so that a simple thermodynamic limit without a surface term is impossible to take. The equilibrium statistical mechanics for the polycondensation was later refined by Yan [77] by taking the surface correction into tree-like systems. He found the same result as Ziff and Stell.

Later, in order to ensure the equilibrium distribution, additional terms describing the reversible reaction (fragmentation) were introduced to the kinetic equation by van Dongen and Ernst [78]. Since their study was limited only to Flory’s and Stockmayer’s model, the possibility of other new treatments within the classical tree statistics for reversible gelation remains an open question. From the mathematical analysis given in this paper, however, it is highly probable that a new thermodynamically consistent treatment, even if it exists, leads to the third-order singularity lying somewhere between Stockmayer’s one and Flory’s one.

Since the appearance of APS, there have been studies accumulated on thermoreversible gelation by computer simulation. They are mainly concerned with the percolation of the clusters, and there have only been a few reports that seriously check the osmotic pressure. Kumar and Panagiotopoulos [79] used a chain model carrying strongly associative stickers regularly placed along the chain, and studied the nature of the transition by a grand canonical Monte Carlo simulation on a lattice. They showed that the osmotic pressure exhibits a cusp-like behavior at a low temperature, just like the form shown in Figure 1 of the present paper. They attributed the behavior to the critical micelle concentration (cmc) because it was independent of the system size, and reached the conclusion that it would not grow to a singularity. Since the third order singularity is expected to be very weak and difficult to detect experimentally, further careful studies on the pressure by computer simulation are eagerly hoped for.

### 2.5. Percolation Models for Gelation

In a quite different way from the classical theory of gelation, the percolation model of gelation focuses on the geometrical structure and connectivity of the system. We describe the percolation model with an attempt to apply it to the gelation problem [70,80,81,82].

Percolation models are roughly classified into percolation on regular lattices and percolation in continuum space. Both of them study the scaling laws for the anomaly of geometrical and physical properties near the percolation threshold on the basis of the self-similarity of the connected objects. In this section, we consider percolation on regular lattices. Percolation in continuum space will be discussed in the following section.

There are two types of lattice percolation problems: site percolation and bond percolation.

#### 2.5.1. Site Percolation

First, we focus on the site percolation. In a site percolation, molecules are randomly distributed on the lattice sites. Neighboring pairs of molecules are regarded as connected. Let Ω be the total number of the lattice sites, and *N* be the number of molecules placed on them. The percolation probability *p* defined by p≡N/Ω is identical to the volume fraction ϕ discussed above. When *p* exceeds a certain threshold value $p_{c}$, connected particles (clusters) of infinite size appear. This critical value $p_{c}$ depends on the space dimensions *d* and the lattice structure.

The cluster distribution function $f_{l}$ is defined by
(98)$$f_{l}\left(p\right)\equiv \frac{N_{l}}{N}$$
wher $N_{l}$ (l=1,2,3⋯) is the number of connected clusters consisting of *l* molecules (referred to as *l*-mers). The number density $\nu_{l}$ of *l*-mers is defined by
(99)$$\nu_{l}\left(p\right)\equiv \frac{N_{l}}{\Omega }$$

In the region $p>p_{c}$ after the percolation threshold is passed, the infinite cluster coexists with finite clusters. Let $N_{\infty }$ be the number of molecules in the infinite cluster. The total molecules are decomposed into two parts
(100)$$N=\sum_{l\geq 1}lN_{l}+N_{\infty }$$

Dividing by Ω, we find the relation
(101)$$p=\sum_{l=1}^{\infty }l\nu_{l}+P_{\infty }\left(p\right)$$
where
(102)$$P_{\infty }\left(p\right)\equiv \frac{N_{\infty }}{\Omega }$$
is the volume fraction of the infinite cluster. The gel fraction $w^{G}$ is then given by
(103)$$w^{G}=\frac{N_{\infty }}{N}=\frac{P_{\infty }\left(p\right)}{p}$$

In the critical region near the percolation threshold, the structure of the clusters are self-similar; the structure observed in a certain length scale looks similar to a part of it when the part is magnified properly, and, hence, they are superimposable to each other. Thus, it is known that the cluster distribution function obeys the scaling law
(104)$$f_{l}\left(p\right)=\frac{1}{l^{\tau }}F\left(\frac{l}{l^{*}\left(p\right)}\right)$$
where τ is a power index referred to as the Fisher index, and $l^{*}\left(p\right)$ is the reference size of the clusters [70,80,81]. The size $l^{*}\left(p\right)$ is shown to be the *z*-average cluster size ${\bar{l}}_{z}$. Practically, it is the size of the largest cluster. Since it diverges at $p_{c}$, the index σ is introduced by the scaling law
(105)$$l^{*}\left(p\right)≃{\left|p-p_{c}\right|}^{-1/\sigma }$$

The index σ and τ are two fundamental structural indices of the percolation theory. The function F(x) is a smooth scaling function that decays sufficiently quickly. On the basis of these two power indices τ and σ, scaling laws of various averages and the gel fraction can be derived [80,81].

The defect of the site percolation model is that all clusters are fixed on the lattice. There is no translational motion of their mass centers, so the pressure and temperature effect cannot be studied. If we assume that clusters obeying the distribution function (Equation 89) move freely with no inter-cluster interaction as in the Frenkel–Band model of associated particles, the pressure is proportional to the total number $\nu^{S}\equiv \sum_{l\geq 1}\nu_{l}$ of finite clusters. Its slope is therefore
(106)$$\left(\frac{\partial \nu^{S}}{\partial \phi }\right)=\frac{1}{{\bar{l}}_{w}}$$
and we go back to Mayer’s Equation (Equation 6). The scaling law with the index τ leads to a singularity of the compressibility.

Another serious deficiency of the site percolation model resides in the effect of temperature. At high percolation probability *p* (volume fraction of the particles), in particular, at the limit of p=1, the system always remains percolated no matter how high the temperature is. In other words, there is no temperature-dependent transition.

#### 2.5.2. Site–Bond Correlated Percolation

To overcome such deficiencies of the site percolation model, Coniglio et al. [83,84] introduced random bond formation between the nearest neighboring molecules in the standard lattice gas model. Solvent (A) and solute (B) molecules placed on a lattice interact with each other in two ways: the usual van der Waals interaction $W_{\alpha ,\beta }$ (α,β= A,B) and reversible bond formation with the bond energy $\epsilon_{BB}$. In what follows, the model is referred to as *site–bond percolation* (SBP). In this paper, we focus only on the bond formation and neglect the van der Waals interaction, as in the preceding sections. Hence, we fix $W_{AA}=W_{AB}=W_{BB}=0$, and $\epsilon_{BB}=0$ with probability $\rho_{u}$ for non-bonded solute pairs, and $\epsilon_{BB}=-\epsilon$ with probability $1-\rho_{u}$ for bonded pairs. The Hamiltonian is
(107)$$H=-\epsilon \sum_{<i,j>}\tau_{ij}\pi_{i}\pi_{j}-\mu \sum_{i}\pi_{i}$$
where $\pi_{i}=0,1$ is the variable for the lattice gas solute molecules, $\tau_{ij}=0,1$ is the variable of the bond formation for the pair (i,j), and μ is the chemical potential of the particle. The grand partition function is then calculated by
(108)$$\Xi \left(T,\mu \right)=\sum_{\left\{\tau \right\}}\sum_{\left\{\pi \right\}}\exp\left\{\beta \epsilon \sum_{<i,j>}\tau_{ij}\pi_{i}\pi_{j}+\beta \mu \sum_{i}\pi_{i}\right\}$$

Summing up with respect to all $\tau_{i,j}$ (annealed average), we find that
(109)$$\Xi \left(T,\mu \right)=\sum_{\left\{\pi \right\}}\exp\left\{K\sum_{<i,j>}\pi_{i}\pi_{j}+L\sum_{i}\pi_{i}\right\}$$
where $K\equiv \ln[\rho_{u}+\left(1-\rho_{u}\right)e^{\beta \epsilon }]$ and L≡βμ. By introducing the Ising variables $\sigma_{i}=\left\{-1,1\right\}$ through the relation $\pi_{i}=\left(1+\sigma_{i}\right)/2$, the grand partition function can be related to the canonical partition function
(110)$$Z\left(J,H\right)\equiv \sum_{\left\{\sigma \right\}}\exp\left\{J\sum_{<i,j>}\sigma_{i}\sigma_{j}+H\sum_{i}\sigma_{i}\right\}$$
for the Ising magnet by the equation
(111)$$\Xi \left(T,\mu \right)=e^{-\Omega (fK/4+L)/2}Z\left(J,H\right)$$
where J≡K/4 and
(112)$$H\equiv \frac{1}{2}\left(\frac{fK}{2}+L\right)=\frac{\beta \mu }{2}+\mathrm{const}$$
(*f* is the number of the nearest neighboring sites). The volume fraction of the solute molecules is given by
(113)$$\phi =\frac{1}{\Omega }\sum_{i}\langle \pi_{i}\rangle =\frac{1}{2}\left[1+m(J,H)\right]$$
where
(114)$$m\equiv \langle \sigma_{i}\rangle =-{\left(\frac{\partial F}{\partial H}\right)}_{J}$$
is the maginitization of the Ising magnet, and $F=-\Omega^{-1}\ln Z(J,H)$ is the free energy of the Ising magnet. The pressure is given by
(115)$$\frac{p}{k_{B}T}=\frac{fK}{8}+H-F\left(J,H\right)$$

Therefore, if we knew the free energy of an Ising ferromagnet as a function of the temperature and the magnetic field, we can study the *p*–*v* curve of the lattice gas.

The slope of the pressure in the isotherm is then obtained by
(116)$${\left(\frac{\partial p/k_{B}T}{\partial \phi }\right)}_{T}={\left(\frac{\partial p/k_{B}T}{\partial H}\right)}_{J}{\left(\frac{\partial H}{\partial \phi }\right)}_{T}$$

Using the definition of the magnetization, we find
(117)$${\left(\frac{\partial p/k_{B}T}{\partial H}\right)}_{J}=1+m=2\phi$$
so that
(118)$${\left(\frac{\partial p/k_{B}T}{\partial \phi }\right)}_{T}={\left(\frac{\partial \ln z}{\partial \ln\phi }\right)}_{T}$$
where $z\equiv e^{\beta \mu }$ is the activity. This relation is in agreement with the relation (Equation 6) in Mayer’s theory, and also Equation (Equation 30) in APS, except the last term. The last term of Equation (Equation 30) comes from the difference between the chemical potential of the solvent in a solution and that of a vacancy of a gas. If we have vacancies instead of solvent molecules, we have, for the activity of the solute molecules,
$$\beta \mu_{1}=\ln z=\frac{1}{n}\left(1+w\frac{\partial }{\partial \ln\phi }\right)\ln\phi_{1}-\ln\left(1-\phi \right)$$

Hence, the two relations are identical., so that it is a model-independent general relation.

In the ideal case where there is no bond formation, we have J=0, so that m=tanhH. The pressure and concentration are given by
(119)$$\frac{p}{k_{B}T}=H+\ln\left(2\cosh H\right)$$
and
(120)$$\phi =\frac{1}{2}\left(1+\tanh H\right)$$

By eliminating *H*, we find
(121)$$\frac{p}{k_{B}T}=-\ln\left(1-\phi \right)$$
and hence
(122)$$\frac{\partial p/k_{B}T}{\partial \phi }=\frac{1}{1-\phi }$$

This agrees with the APS Equationn (Equation 48), with n=1 and ${\bar{l}}_{w}=1$.

#### 2.5.3. Exact Solution on the Bethe Lattice

We next consider the exact solution of SBP on the Bethe lattice presented by Coniglio et al. [83,84]. It is well known that the Ising model of a ferromagnet on a Bethe lattice can be solved exactly. The result is summarized for instance in Baxter [67]. The solution was applied to the site percolation problem of the Ising lattice gas by Odagaki [85] and Coniglio [86]. We first review these works on the site percolation problem, and then go back to the SBP problem.

The solution of the Ising ferromagnet on a Bethe lattice can be described as follows. Let $\xi \equiv e^{-2J}$, $\eta \equiv e^{-2H}$, and $\eta_{1}\equiv \eta x^{f-1}$ be the three fundamental parameters. These parameters are written as $z\equiv e^{-2K}$, $\mu \equiv e^{-2h}$ and $\mu_{1}\equiv \mu x^{q-1}$ in Ref. [67]. Since we use the letter *z* for the activity and μ for the chemical potential, we have changed these letters. The coordination number *q* is replaced by *f* to compare with the Flory–Stockmayer theory. The parameter *x* is defined by
(123)$$x\equiv \frac{\xi +\eta_{1}}{1+\xi \eta_{1}}$$
and is related to the magnetization by
(124)$$m=\frac{1-\eta x^{f}}{1+\eta x^{f}}$$

In terms of the activity $z\equiv e^{L}=e^{\beta \mu }$, we find $\eta =\xi^{f}/z$ because H=(fK/2+L)/2. By eliminating η and $\eta_{1}$, we find
(125)$$z=\frac{\xi (1-\xi x)}{x-\xi }{\left(\xi x\right)}^{f-1}$$

The concentration of the solute molecules is
(126)$$\phi =\frac{1}{2}\left(1+m\right)=\frac{1}{1+\eta x^{f}}=\frac{z}{z+{\left(\xi x\right)}^{f}}$$
and the pressure can be found from the exact solution of the Ising free energy [67] as
(127)$$\frac{p}{k_{B}T}=\frac{f}{2}K+\frac{1}{2}\ln z+\frac{f}{2}\ln\left(1-\xi^{2}\right)-\frac{1}{2}\ln\left\{1+\xi^{2}-\xi \left(x+\frac{1}{x}\right)\right\}-\frac{1}{2}\left(f-2\right)\ln\left(x+\frac{1}{x}-2\xi \right)$$

We then eliminate the activity *z* by using Equations (Equation 115) and (Equation 116), and find *x* in terms of the concentration and temperature as [85]
(128)$$x=\frac{\xi }{2\phi }\left\{-\left(1-2\phi \right)+\sqrt{1+4\phi \left(1-\phi \right)\left(\xi^{-2}-1\right)}\right\}$$

The activity as a function of the concentration is found by substituting this into Equation (Equation 115). With all of these results, we can find the pressure explicitly as a function of the concentration
(129)$$\frac{p}{k_{B}T}=\frac{f}{2}\ln\left\{\frac{1}{2\xi^{-2}}\left[1+2\left(1-\phi \right)\left(\xi^{-2}-1\right)+\sqrt{1+4\phi \left(1-\phi \right)\left(\xi^{-2}-1\right)}\right]\right\}-\ln\left(1-\phi \right)$$

We focus on a solute molecule and find the probability *q* for one of its nearest neighboring cites to be occupied by a solute molecule. It is given by the correlation function $q=\langle \pi_{0}\pi_{1}\rangle /\langle \pi_{0}\rangle$ and calculated to be [85,86]
(130)$$q=\frac{1}{1+\xi \eta_{1}}=\frac{1-\xi x}{1-\xi^{2}}$$

Substituting the above *x*, we find
(131)$$q=\frac{1+2\left(\xi^{-2}-1\right)\phi -\sqrt{1+4\phi \left(1-\phi \right)\left(\xi^{-2}-1\right)}}{2(\xi^{-2}-1)\phi }$$

In this result, we consider the limit of strong bond energy ξ→0 with a fixed $\xi^{-2}\phi$. The factor (1−ϕ) in the square root can be replaced by 1 in this limit, so the probability *q* agrees with the reactivity α in Equation (Equation 56) for APS. The association constant fλ corresponds to the factor $\xi^{-2}-1$ of the Ising model. Because $\xi^{-2}-1=e^{K}-1=\left(1-\rho_{u}\right)\left(e^{\beta \epsilon }-1\right)$, the relation is summarized as
(132)$$f\lambda \left(T\right)=\left(1-\rho_{u}\right)\left(e^{\beta \epsilon }-1\right)$$

In the limit of strong bond energy, 1 is neglected in the second factor. The relation reduces to
(133)$$f\lambda \left(T\right)=\left(1-\rho_{u}\right)e^{\beta \epsilon }$$

Therefore, we see that, if we take the strong bond limit ξ→0 with finite $\xi^{-2}\phi$, the site percolation model reduces to APS with association constant (Equation 123).

The volume fraction of the particles can be written as
(134)$$\phi =\frac{q}{1+{\left(1-q\right)}^{2}\left(\xi^{-2}-1\right)}$$
in terms of the probability *q*. This equation reduces to APS Equation (Equation 55) in the limit of the strong bonds (ξ<<1).

The weight-average molecular weight of the connected clusters in the site percolation problem is given by [85,86]
(135)$${\bar{l}}_{w}\left(x\right)=\frac{1+q}{1-(f-1)q}=\frac{2-\xi (x+\xi )}{1-\xi^{2}-\left(f-1\right)\left(1-\xi x\right)}$$
so that the percolation point is decided by the equation
(136)$$1-\xi^{2}-\left(f-1\right)\left(1-\xi x\right)=0$$
or q=1/(f−1). The concentration at the percolation point is therefore
(137)$$\phi^{*}=\frac{f^{\prime }}{2f-3+{f^{\prime \prime }}^{2}\xi^{-2}}$$
where $f^{\prime }\equiv f-1,f^{\prime \prime }\equiv f-2$, etc., have been used.

We now see the trouble with this site percolation model. In the limit of high temperature ξ→1, the percolation line has a finite limiting value $\phi^{*}=1/f^{\prime }$. In other word, the system remains percolated no matter how high the temperature is if the concentration is higher than the critical value $1/f^{\prime }$. There is no percolation transition by the temperature change. This unphysical result originates from the assumption that a nearest neighboring pair is always regarded as connected. In fact, q=1 (connectivity 1) always trivially holds at ϕ=1 (no solvent) as is seen from Equation (Equation 121).

To remedy this defect of site percolation, Coniglio et al. [83,84] introduced the SBP model, in which, the nearest neighboring pair is either bonded (with probability $p_{B}$) or unbonded (with probability $1-p_{B}$). The connectivity probability *q* is replaced by
(138)$$\alpha \equiv p_{B}q$$
in all of the above relations. In particular, the weight-average molecular weight and the percolation threshold is given by
(139)$${\bar{l}}_{w}=\frac{1+\alpha }{1-(f-1)\alpha }$$
as in Equation ([Disp-formula FD11334-ijms-23-10325]), but now $\alpha =p_{B}q$.

This relation is the same both in APS (the limit of strong bond) and SBP (bond formation introduced for the nearest neighboring pairs). Both models have a sol–gel transition at full volume fraction ϕ=1. However, the thermodynamic nature of the transition is different.

In order to see the difference, we study the slope (Equation 108) of the pressure. First, by taking the derivative of Equation (Equation 115) with respect to the concentration, we find
(140)$$\frac{\partial \ln x}{\partial \ln z}=\frac{1}{f^{\prime }-\left(1-\xi^{2}\right)x/\left(1-\xi x\right)\left(x-\xi \right)}$$

Substituting into the relation
(141)$$\frac{\partial \ln\phi }{\partial \ln z}=\left(1-\phi \right)\left(1-f\frac{\partial \ln x}{\partial \ln z}\right)$$
obtained from the derivative of Equation (Equation 116), we find
(142)$$\frac{\partial p/k_{B}T}{\partial \phi }=\frac{\kappa (x)}{1-\phi }$$
where
(143)$$\kappa \left(x\right)\equiv \frac{\left[1-\xi^{2}-f^{\prime }\left(1-\xi x\right)\right]x+f^{\prime }\left(1-\xi x\right)\xi }{\left(1-\xi^{2}\right)x-\left(1-\xi x\right)\left(x-\xi \right)}$$

The spinodal condition is therefore given by κ(x)=0, or, explicitly,
(144)$$\left(1-\xi^{2}\right)x-f^{\prime }\left(1-\xi x\right)\left(x-\xi \right)=0$$

This condition for the spinodal of the Ising model is different from the percolation condition (Equation 126), and hence the percolation line is not accompanied by any singularity. However, if we take the strong bond limit ξ→0 in this spinodal condition (with finite ξx), we can see that the spinodal condition becomes identical to the percolation condition. In other words, the slope of the pressure vanishes on the percolation line.

As for the treatment of the post-gel regime, the percolation model on a Bethe lattice has a serious problem. A Bethe lattice with a finite number *N* of cites has the total fN−2(N−1) of the free functional groups on the surface if we fill the lattice with functional molecules. Hence, in the thermodynamic limit of N→∞, the reactivity of the nearest neighboring molecular pairs has an upper limit 2/f. Therefore, Flory’s treatment allowing for cycle formation is impossible. This is one of the most important differences between the classical tree statistics of the polyfunctional molecules in a three-dimensional off-lattice free space and the percolation model on a Bethe lattice. For the latter, Stockmayer’s treatment is the only possible treatment of the post-gel regime.

### 2.6. Adhesive Hard Sphere Model

Let us move onto percolation in a continuum space. In the study of the gas–liquid phase transition of spherical particles, Baxter introduced a model in which hard sphere particles interact with each other by the attractive force potential of a rectangular well shape (referred to as an adhesive hard sphere model AHS; see Figure 5a,b) [87,88]. In the limit of the surface adhesion, i.e., narrow force range and strong attractive force, he found the analytic solution of the thermodynamic problem within Percus–Yevick approximation. Since then, AHS is often used as a model system for the study of gelation phenomena in globular proteins, colloid dispersions, silica aerogels, and other particulate gels.

Consider *N* spherical particles of radius σ/2 in a container of volume *V*. The volume fraction $\phi =\left(4\pi /3\right){\left(\sigma /2\right)}^{3}N/V$ is used for the concentration. The attractive potential u(r) has a depth −ϵ and width Δ≡d−σ. Baxter [87] introduced the reduced temperature τ by the definition
(145)$$\frac{\epsilon }{k_{B}T}\equiv \ln\left[12\tau \left(1-\frac{\sigma }{d}\right)\right]$$

The second virial coefficient of AHS is
(146)$$A_{2}\left(T\right)=-\frac{1}{2}\int_{0}^{\infty }f\left(r\right)4\pi r^{2}dr$$
where
(147)$$f\left(r\right)\equiv e^{-u\left(r\right)/k_{B}T}-1=\frac{\sigma }{12\tau }\delta \left(r-\sigma \right)+\theta \left(r-\sigma \right)-1$$
is the Mayer’s function (Equation 2) for AHS. It can be normalized as $A_{2}^{*}\equiv A_{2}\left(T\right)/A_{2}^{\mathrm{HS}}$ by using the hard sphere system $A_{2}^{\mathrm{HS}}=\left(4\pi \sigma^{3}/3\right)$ as the reference system. In the Baxter limit, $\tau =1/4(A_{2}^{*}-1)$. If we apply Mayer’s theory of condensation, the coefficients (cluster integrals) $b_{l}$ for AHS are constructed by this special form of f(r).

In the limit of short range d→σ and strong force ϵ→∞, the spheres form *branched sponge-like clusters* in which they are connected to each other by surface adhesion [87]. Above a certain volume fraction, the clusters percolate over the entire container [89,90,91,92].

AHS systems show interesting phase diagrams in which gas–liquid phase transition (thermal problem) coexists with percolation transition (connectivity problem). The percolated cluster can be regarded as a porous gel comprising spherical particles. There have been theoretical attempts to construct phase diagrams on the τ-ϕ plane [90,91]. Within Percus–Yevick approximation, Chiew and Glandt [90] found that the percolation line is given by In the dilute region, this approximation is poor. In particular, the gelation temperature must decrease to −∞ in the limit of ϕ→0, while Equation (Equation 148) gives a finite τ.
(148)$$\tau =\frac{19\phi^{2}-2\phi +1}{12{\left(1-\phi \right)}^{2}}$$

Because the analytical solution of the problem is difficult to find, molecular simulations are used to construct the phase diagram [93,94,95,96,97].

From a thermodynamic point of view on the sol–gel transition, AHS retains the same deficiency as the site percolation; in the limit of high density, the system remains percolated at any high temperature. To refine AHS, there have been several attempt to replace the Baxter potential by patchy sticky hard spheres [98,99,100], which carry a varied number *f* of attractive patches on the surface (Figure 5c). Two spheres interact via a sticky Baxter potential if the line joining the centers of the two spheres intersects a patch on each sphere, and via a hard sphere potential otherwise. The area and distribution of attractive patches on the sphere surface are changed to study how the percolation line and phase separation (gas–liquid) line shift. The analytical study and Monte Carlo simulation [98] on the systems with f=1,2 with a varied patch area showed that the percolation line shifts to a higher concentration and lower temperature region as the patch area decreases. However, because the analytical study of the pressure was based on the virial expansion up to the third order of the density, or on the integral equation approximation for closure, it was impossible to find any singularity across the percolation line. If the number *f* is increased, with a smaller patch area, the cluster integrals of such a patchy AHS take the form similar to $\omega_{l}$ in tree statistics with the functionality *f* (Figure 5d) for a large cluster size *l*. There is a strong tendency to form tree-type clusters rather than spherical droplets. As a result, the percolation transition splits substantially away from the gas–liquid transition line, as was pointed out by Stillinger [43] for the association model. We can expect that patchy AHS with f≥3 reveals a temperature-controlled transition even in the limit of a high density.

## 3. Discussion

As for the absolute value of the discontinuity in the slope of the osmotic compressibility, the experimental observation seems to be difficult to detect since the amplitude *B* is small for polymeric gels with n>>1. (For a small *f*, such as trifunctional molecules with f=3, for example, we have $B=9\lambda^{4}/8n^{5}$.) However, there is a good chance for observation to be successful if we take advantage of the interference of gelation with phase separation. In most associating solutions, molecules interact not only with associative force but also non-associative vdW force. The latter (positive χ-parameter) leads to liquid–liquid phase separation with the spinodal condition (Equation 35) with modified Equation (Equation 36). As we approach the spinodal point along the sol–gel transition line where the condition $\sigma (\phi^{*},T)=0$ is satisfied, the discontinuity in the compressibility is enhanced by critical fluctuations, as seen from Equation (Equation 73), and there may be a chance to observe the singularity.

The effect of fluctuations in cluster formation is important near the sol–gel transition point as it usually is in many phase transitions. It may change the nature of the transition. In the present gelation problem, one has to consider it from two sides. Firstly, the effect of cycle formation during the gelation reaction beyond the classical tree statistics must be considered. Secondly, concentration fluctuations in the polymer solution near a critical point must be considered beyond the mean-field Flory–Huggins treatment. These two different aspects of fluctuations should be studied from a unified theoretical point of view without violating the fundamental law of thermodynamics. This is an open problem. The critical exponent and the nature of the singularity may be changed by these fluctuation effects near the transition point. If this is the case, the criterion necessary for the mean-field prediction to be valid (Ginzburg criterion) should also be found in order to observe non-classical singularity in the experimeriments.

## 4. Conclusions

We have shown that, within the mean-field treatment of both the gelation reaction and polymer solution (the model of associating polymer solutions), thermoreversible gelation is a third-order phase transition analogous to Bose–Einstein condensation. The gel network corresponds to a Bose condensate with no translational motion (no momentum). There is a frequent exchange of molecules between the gel network and the sol part (particles with finite momentum), but, on thermal average, a finite fraction of the number of molecules lose their momenta in the post-gel regime. Physically, this is caused by the sparse surface structure of associated (cross-linked) particles (clusters) with no surface tension. The site–bond percolation model on a Bethe lattice leads to the same conclusion if we take the limit of a strong associative force.

In order to avoid complexity and focus on gelation, we considered associative force only, and discarded the usual van der Waals (vdW) interaction by which the solution is driven into the liquid–liquid phase separation. We may construct phase diagrams showing phase separation coexisting with sol–gel transition by the inclusion of vdW force in terms of conventional χ-parameters. There are several types of the interference of the two phase transitions depending on the balance between the vdW force and associative force. The sol–gel transition line crosses the phase separation line (binodal and spinodal) at the top, or on the shoulder, of the phase separation region [101]. Such a crossing point is called either a *tricritical point* or *critical endpoint* depending upon the relative position of the transition curves.

## Figures and Tables

**Figure 1 ijms-23-10325-f001:**
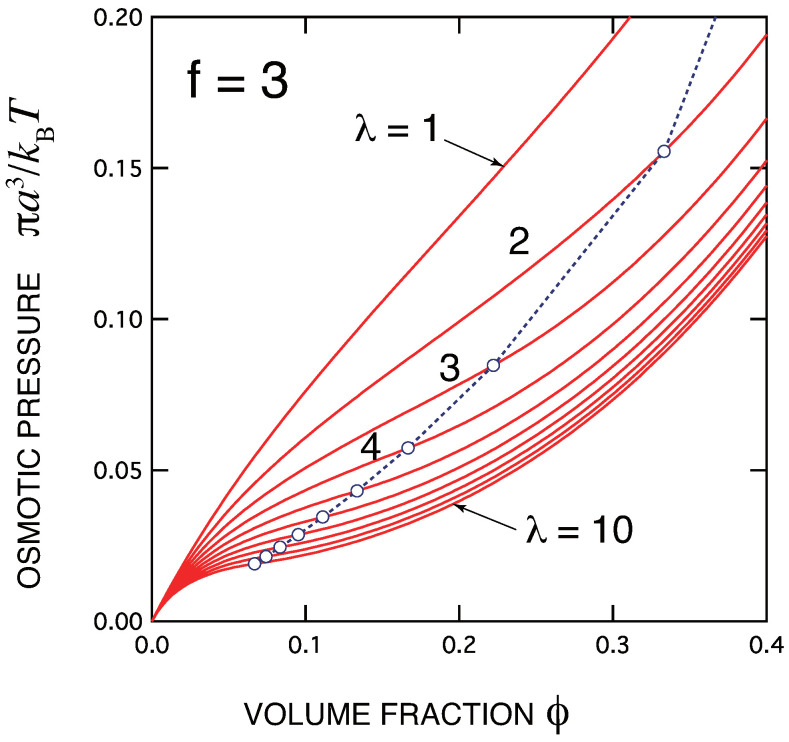
The osmotic pressure $\pi a^{3}/k_{B}T$ plotted against the volume fraction of the primary molecules of low molecular weight (n=1). The functionality of the molecule is fixed at f=3. The association constant λ is varied from curve to curve. The gel points are indicated by circles.

**Figure 2 ijms-23-10325-f002:**
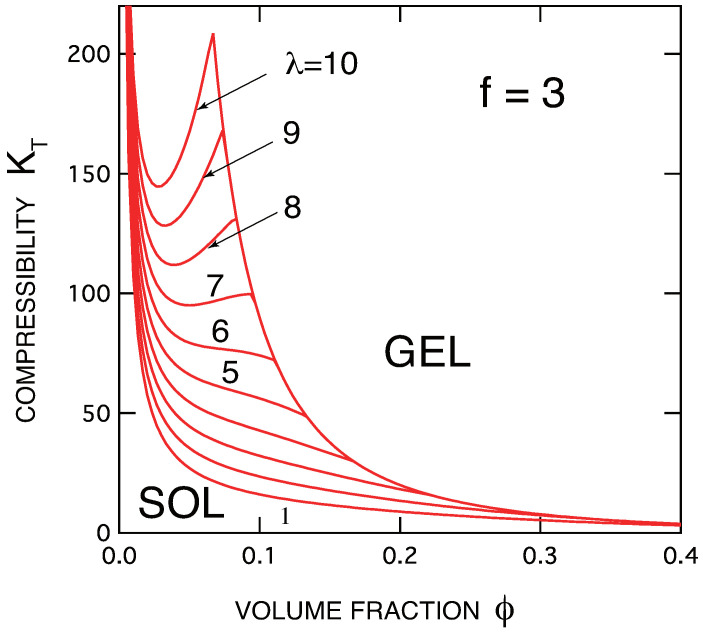
The osmotic compressibility $K_{T}\equiv \left(\partial \phi /\partial \left(\pi a^{3}/k_{B}T\right)\right)/\phi$ plotted against the volume fraction of the primary molecules of low molecular weight (n=1). The functionality of the molecule is fixed at f=3. The association constant is varied from curve to curve. It is continuous across the gel point, but its concentration derivative is discontinuous.

**Figure 3 ijms-23-10325-f003:**
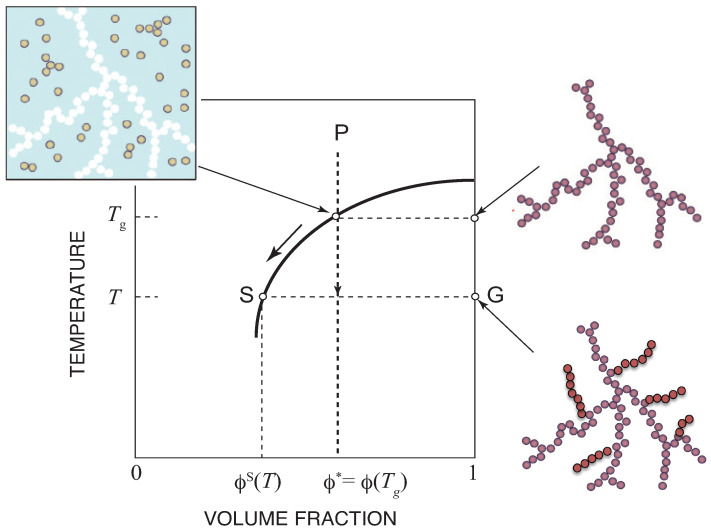
Sol–gel equilibrium in Stockmayer’s post-gel treatment shown on the temperature–concentration plane. The entire solution contains the sol part (S) and the gel part (G) in equilibrium in a spatially homogeneous state. By cooling, the concentration of the sol remains at the critical value so that it moves along the sol–gel transition line.

**Figure 4 ijms-23-10325-f004:**
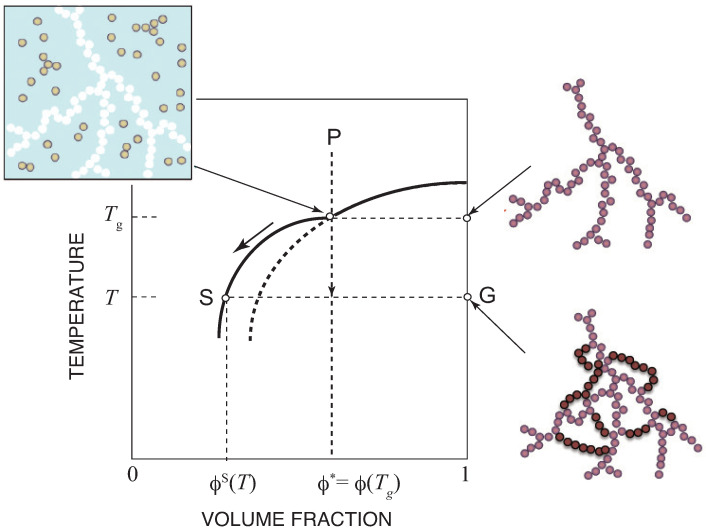
Sol–gel equilibrium in Flory’s post-gel treatment shown on the temperature–concentration plane. By cooling, the concentration of the sol deviates from the sol–gel transition line to lower concentration region because the concentration of the sol part becomes lower than the critical value as gelation proceeds. The gel parts contains cycles.

**Figure 5 ijms-23-10325-f005:**
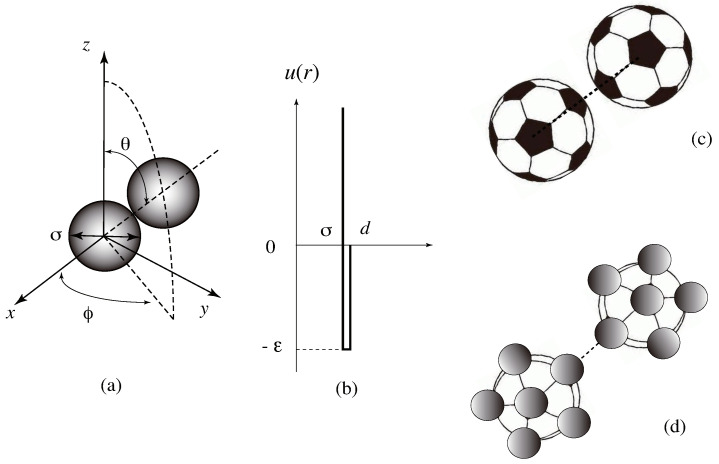
(**a**) Interacting adhesive hard spheres, (**b**) Baxter potential, (**c**) patchy hard spheres iteracting by adhesive Baxter force, (**d**) functional molecules employed by Flory–Huggins gelation model. Functional groups (dark spheres) on the surface of molecules interact by adhesive Baxter force.

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
