# Peer review of "Comparative Study on the Models of Thermoreversible Gelation"

_ijms, 2022, doi:10.3390/ijms231810325_

Round 1

Reviewer 1 Report

The present manuscript appears as having only one author, however in the text "we" is often used;

The manuscript brings several theoretical models of the thermoreversible gelation, but the proposed discussion is very simple; since it is the main proposal of the article, it should be improved!

I strongly recommend a review by a native English speaker.

Author Response

Reviewer: 1

Thank you for reading the paper for review. My incorporation of your suggestions is as follows.

(1) The discussion presented here on the thermoreversible gelation is already very complex. The models proposed so far in the past literature are so diverse that I needed many pages to clarify their relationship. I think the present volume and style of the paper is appropriate for the readers, in particular experimentalists working in this field,  to understand the status of the topic.

(2) I have changed “We show …” to “I show …” in the abstract according to your suggestion. In the main text, however, I retained the style of “we…” because of the two reasons:

1) Most of the parts refer to the past works I have done with my research collaborators. I would like to include them whenever I write on the results.

2) I would like to include readers of my paper by writing “Let us …”, “ We go back to…” We assume …” etc. I think it is very natural English style of scientific papers.

(3) Some sentences are grammatically corrected to improve English. Several typos are corrected. They are highlighted in the .pdf file.

I hope the manuscript has been satisfactorily improved and it now warrants publication in IJMS

Reviewer 2 Report

The paper is very good critical survey on different theoretical models of the thermoreversible gelation. The Author is a prominent specialist in the area. I think that the paper could be printed in the presented form. I have only several remarks on the form and edition.

Page 47 and 48...why Discussion after Conclusions?

(67) Reference,30 Chap.14

(60) Reference,24 Chap. XII

Why not to cite reference 30 or 24?

(78) In the literature (A.N. Semenov and M. Rubinstein, Macromolecules, 1998, 31, 1373-1385), there is a statement that "In fact one can easily check that the free energy, eqn (2.15), and all its derivatives are perfectly analytical at the gel point." The error in this paper resides in the treatment of the reaction in the postgel regime. It claims that the study is based on Flory’s postgel picture, but in fact it simply missed the extent of reaction α′ of the sol, and hence it failed to find the gel fraction.

...and in several other literature positions...

Is it acceptable to make remarks in the references?

Author Response

Reviewer: 2

Thank you for reading the paper for review. My incorporation of your suggestions is as follows.

(1) References (60),(67) are eliminated. Instead, in the text,  “Chapter XII of Flory’s textbook^24”  etc are written. They are highlighted.

(2) Four comments are eliminated from the reference, and moved to the bottom of the text pages.

(3) I have suggested in the “Discussion” section the potential method of experimental investigation on the conclusions obtained in this paper. I have also referred to the limitation of our theory (mean field approximation) in the “Discussion”.  It is natural that these are written after “Conclusion” section.

(3) Some sentences are grammatically corrected to improve English. Several typos are corrected. They are highlighted in the .pdf file.

I hope the manuscript has been satisfactorily improved and it now warrants publication in IJMS

Round 2

Reviewer 1 Report

The author has improved the points commented on in the review. I recommend the present work for publication.